# Construction, Physical Properties and Foaming Behavior of High-Content Lignin Reinforced Low-Density Polyethylene Biocomposites

**DOI:** 10.3390/polym14132688

**Published:** 2022-06-30

**Authors:** Seo-Hwa Hong, Seok-Ho Hwang

**Affiliations:** Materials Chemistry & Engineering Laboratory, Department of Polymer Science & Engineering, Dankook University, Yongin 16890, Gyeonggi-do, Korea; seohwa8766@naver.com

**Keywords:** lignin, low-density polyethylene, biocomposites, chemical modification, foamability

## Abstract

Lignin was chemically modified with oligomeric polyethylene (oPE) to form oPE-grafted lignin (oPE-*g*-lignin) via lignin surface acylation and a radical coupling reaction with oPE. Then, pristine lignin and oPE-*g*-lignin were successfully compounded with low-density polyethylene (LDPE) through a typical compounding technique. Due to the oligomeric polyethylene chains grafted to the lignin’s surface, the interfacial adhesion between the lignin particles and the LDPE matrix was considerably better in the oPE-*g*-lignin/LDPE biocomposite than in the pristine-lignin/LDPE one. This demonstrated that oPE-*g*-lignin can serve as both a biodegradable reinforcing filler, which can be loaded with a higher lignin content at 50 wt-%, and a nucleating agent to increase the crystallization temperature and improve the tensile characteristics of its LDPE biocomposites. Moreover, the foamability of the lignin-reinforced LDPE biocomposites was studied in the presence of a chemical blowing agent (azodicarbonamide) with dicumyl peroxide; for an oPE-*g*-lignin content up to 20 wt-%, the cell size distribution was quite uniform, and the foam expansion ratios (17.69 ± 0.92) were similar to those of the neat LDPE foam (17.04 ± 0.44).

## 1. Introduction

Polyolefin-based foams are among the most important polymeric foams used since the early 1940s. Their commercial application has expanded rapidly to various industrial areas, such as construction, transportation, sport and leisure, packaging, and agriculture [1,2,3]. Their success in so many fields is because of their unique properties, including a low density and water absorption, a high strength-to-weight ratio, and good thermal insulation, chemical resistance, and cushioning performance [3,4,5].

Despite the numerous advantages of polyolefin-based foams, their non-recyclability and nonbiodegradability significantly limit their widespread usage due to serious plastic-related environmental problems [6]. To overcome such fatal drawbacks for polyolefin-based foams, biobased or biodegradable polymeric systems and their fabrication technologies have been continuously developed as an alternative to fully petroleum-based polyolefins [7,8,9,10,11].

As a possible solution for the aforementioned issues, blending polyolefins with biodegradable or biomass-based nondegradable polymers has been adopted to obtain eco-friendly polyolefins [12,13,14,15]. However, most biopolymers produced with this technique, unfortunately, are immiscible with polyolefins, which drastically weakens their mechanical properties due to their physical and chemical differences. Polyolefin-based biocomposites have so far attracted extensive attention as an alternative approach [16,17,18,19,20]. Among their numerous biodegradable fillers, lignin is an important candidate because it is an inexpensive, natural, and biodegradable material obtained during the pulping and biorefinery processes [21,22,23,24,25]; nonetheless, since polyolefins and the lignin filler are still immiscible, chemical and physical methods have been proposed to modify the characteristics of lignin to enhance its dispersibility and interface adhesion in a two-phase composite system [26,27,28,29,30,31].

Since there are tremendous results for low-density polyethylene (LDPE)-based composites with various fillers [32,33,34,35], based on these preliminary results, the primary objective of the present study was the fabrication of chemically modified lignin particles with an improved interfacial adhesion to low-density polyethylene (LDPE) and the successive evaluation of LDPE biocomposites with a lignin content up to 50 wt-%. Thereafter, the foamability of such lignin-reinforced LDPE biocomposites in the presence of a traditional chemical blowing agent was investigated, since many previous works have reported the fabrication of PE foams with varied crosslinking and blowing agent concentrations [36,37,38,39,40,41]; in particular, the effect of the chemically modified lignin on their thermal/physical properties and foamability was determined.

## 2. Materials and Methods

### 2.1. Materials

Low-density polyethylene (LDPE) (LUTENE FB0800; melt flow index: 0.8 g/10 min; density: 0.921 g/cm^3^) was purchased from LG Chem (Seoul, Korea). Lignin (organosolv type; pH = 6.9~7.1; ash < 16%) was obtained from BOC Sciences (Shirley, NY, USA). Oligomeric polyethylene wax (m.p. = 109–111 °C; density: 0.95 g/cm^3^) and acryloly chloride were purchased from Micro Powder, Inc. (Tarrytown, NY, USA) and Jihyun Chem. (Yongin, Korea), respectively. Azodicarbonamide (ADA), zinc oxide (ZnO), and dicumyl peroxide (DCP) were purchased from Sigma-Aldrich (St. Louis, MO, USA). Solvents were purchased from SAMCHUN Chemicals (Seoul, Korea).

### 2.2. Chemical Modification of Pristine Lignin

First, pristine lignin (15 g) and triethylamine (3.5 g) were dispersed in *N*,*N*-dimethylacetamide (DMAc: 200 mL) and stirred for 3 h at 25 °C under N_2_ gas. The mixture was cooled down in the ice bath and then acryloyl chloride (10 g) diluted by DMAc was dropped slowly into the mixture. After completing the dropping, the resultant mixture was stirred for additional 3 h in the ice bath and then stirred for 4 h at 40 °C. The mixture was immediately poured into saturated NaHCO_3_ aqueous solution, then filtered, washed with deionized H_2_O several times, and then kept in vacuo at 80 °C for 1 day to dry completely. Next step, after the oPE (24 g) was completely dissolved in xylene (300 mL) at 120 °C, acrylate-decorated lignin (Ac-Lignin: 4 g) was added into the solution and stirred for 1 h, and then the radical initiator, dicumyl peroxide (1 g), was added. After stirring the mixture for a further 4 h, the mixture was immediately hot filtered, washed with hot xylene solvent several times, and then kept in vacuo at 80 °C for 1 day to dry completely.

### 2.3. Preparation of Lignin-Reinforced LDPE Biocomposites

Using the batch-type internal mixer (RheoComp system, MKE, Deajeon, Korea), the pristine-lignin/LDPE and oPE-*g*-lignin/LDPE biocomposite samples were prepared via compounding technique. LDPE and lignin filler’s various mass ratios were 90/10, 80/20, 70/30, 60/40, and 50/50. The compounding was carried out for 8 min at 140 °C and a rotation speed of 50 rpm.

### 2.4. Foaming of Lignin-Reinforced LDPE Biocomposites

After completely melting the lignin-reinforced LDPE biocomposite samples in the batch internal mixer at 135 °C, ADA (8.0 phr), ZnO (3.0 phr), and DCP (1.2 phr) were then added together and mixed for 4 min. Following a single-stage foaming procedure (compression molding), the foam samples were prepared at 200 °C. The pretreated lignin-reinforced LDPE biocomposite sheet was charged in the house-designed metal closed-mold (100 × 100 × 3 mm), placed on the platen of a hydraulic hot press (Bautek, Pocheon, Korea), and subjected to 60 MPa of pressure for 3 min. Then, the mold was immediately opened to release the pressure.

### 2.5. Characterization

Fourier transform infrared (FT-IR) spectra were collected by the attenuated total reflectance technique with a ZnSe crystal at a nominal incident angle of 45° and a refraction index of 2.4 on the Nicolet *i*S10 spectrophotometer (Thermo Scientific Co., Waltham, MA, USA). The thermal characteristics were determined by differential scanning calorimetry (DSC) on the DSC Q20 apparatus (TA Instruments, New Castle, DE, USA). The data were collected with a heating rate of 20 °C/min under a N_2_ gas atmosphere. TGA/SDTA 851e thermogravimetric analyzer (TGA; Mettler Toledo, Greifensee, Switzerland) was used for measuring the thermal stability under a heating rate of 20 °C/min in a N_2_ gas atmosphere. The tensile properties for lignin-reinforced LDPE biocomposite sample without foaming components (ADA, DCP, and ZnO) were evaluated by Instron universal testing machine (model 6800; Instron, Norwood, MA, USA) according to standard ASTM D638-10 at 25 °C. The crosshead speed was 5 mm/min and the tensile characteristics were calculated from the averages of at least five parallel tests. The fractographic characterization was performed by S-4700 scanning electron microscope (FE-SEM: Hitachi High-Tech, Tokyo, Japan) operated at an accelerating voltage of 20 kV. The apparent density for lignin-reinforced LDPE biocomposite foam was evaluated according to standard ASTM D1622. The cell density (i.e., the number of cells per cubic centimeter) was calculated from the previously reported equation [42] based on SEM images.

## 3. Results and Discussion

Due to its unique characteristics, pristine lignin can hardly exhibit a well-dispersed texture when compounded with synthetic hydrophobic polymers. Thus, to be used as a reinforcing filler in polymeric composites, lignin must be chemically modified; so far, its hydrophilic hydroxyl groups have been crucial in the chemical modification of the lignin’s surface [30,43,44]. In the present study, polyolefin-chain-grafted lignin was constructed via a typical radical reaction (Figure 1); this reaction is completed by an alkene addition reaction of an oligomeric polyethylene (oPE) radical and a coupling reaction between each radical generated through electrophilic addition and hydrogen abstraction of the initiator in the propagation step of the radical mechanism, respectively. For this organic chemistry approach, acrylic monomers and polyethylene oligomers were selected as the lignin’s surface modifiers. We first introduced the acryl groups on the lignin’s surface through esterification, followed by a radical reaction, assuming that the polyethylene component was grafted to the lignin molecules.

Then, to confirm the change in the chemical structure of the oPE-grafted lignin (oPE-*g*-lignin), Fourier-transform infrared (FT-IR) spectroscopy was applied. Figure 1 compares the FT-IR spectra of pristine lignin, lignin esterified with acryloyl chloride (Ac-lignin), and oPE-*g*-lignin. The FT-IR spectrum of pristine lignin adequately agreed with those reported in previous studies [45,46,47,48,49]: a typical broad band due to the phenolic and aliphatic hydroxyl groups in lignin was detected at ~3400 cm^−1^, two distinct peaks were observed at 2925 and 2848 cm^−1^ corresponding to the C–H vibration and C–H stretching of the methoxy groups, respectively, and the aromatic skeleton vibration of lignin resulted in signals at 1599, 1513, and 1460 cm^−1^. After the esterification, two new bands appeared at approximately 1727 and 1122 cm^−1^, suggesting the successful formation of acrylate-decorated lignin including ester groups (one C=O bond and two C–O bonds). As for oPE-*g*-lignin, most of its absorption bands were similar to those of Ac-lignin, but the intensities of the peaks at 2914 and 2847 cm^−1^, attributed to methylene/methine moieties, drastically increased; this means that oPE was successfully introduced on the lignin’s surface through a radical reaction.

The effect of introducing pristine lignin or oPE-*g*-lignin as a reinforcing filler on the thermal behavior and thermal stability of the LDPE matrix were monitored by differential scanning calorimetry (DSC) and thermogravimetry analysis (TGA), and the results are summarized in Table 1. The percent crystallinity ($\chi_{c}$) of the LDPE phase was calculated as
$$\chi_{c}=\frac{\Delta H_{f}}{\Delta H_{f}^{o}}\times 100$$
where $\Delta H_{f}^{o}$ is the heat of fusion for 100% crystalline LDPE (287.6 J/g [50]). The addition of pristine lignin did not change the melting peak temperature (*T*_m_) and non-isothermal crystallization peak temperature (*T*_c_) compared with neat LDPE, which only slightly ranged from 113.4 °C to 114.5 °C and 89.4 °C to 90.1 °C, respectively, and this may be due to the poor interfacial adhesion between pristine lignin and the LDPE matrix. As for the oPE-*g*-lignin/LDPE biocomposites, the *T*_m_ did not substantially change (113.1 °C–114.8 °C), but the *T*_c_ slightly increased compared to neat LDPE. Such a tendency generally occurs when the domain material is compatible with the matrix polymer in a blend or composite system [51,52]. Thus, this indicates that oPE-*g*-lignin could serve as both a reinforcing filler and a nucleating agent in LDPE biocomposites. The TGA thermograms for neat LDPE and their lignin biocomposites were plotted in the figure for comparison (Appendix A). For neat LDPE, the weight loss is at 465 °C, where an 80% weight loss occurred. For pristine lignin (50 wt-%), the weight loss was at 457 °C while for oPE-*g*-lignin (50 wt-%) the weight loss was detected at 469 °C. This indicated that the chemically modified lignin had a better thermal stability than the pristine lignin, which is important from the standpoint of the compounding process.

The dispersibility and interfacial adhesion of a filler in a composite matrix should significantly influence the mechanical properties of filler-reinforced composites. To evaluate how the chemical modification of pristine lignin can affect the dispersibility and the interfacial adhesion between lignin particles and an LDPE matrix, the morphology of cryofractured lignin-reinforced LDPE biocomposites was investigated using scanning electron microscopy (SEM) (Figure 2). The SEM micrographs of the fracture surface of the LDPE biocomposites reinforced with various lignin contents revealed a typical incompatible composite showing several gaps and holes left by lignin particle agglomeration due to a lack of adhesion between the lignin particles and LDPE matrix. As the compatible counterpart, the oPE-*g*-lignin/LDPE biocomposites exhibited a smoother fracture surface, and the lignin was more closely attached to the LDPE matrix; such a morphology demonstrates an improved interfacial adhesion between the oPE-*g*-lignin particles and the LDPE matrix, as evidenced by the presence of oPE chains grafted onto the lignin’s surface.

Figure 3 illustrates the effect of the lignin content on the mechanical properties of the LDPE biocomposites. For the pristine-lignin/LDPE biocomposites, the tensile strength gradually decreased when increasing the lignin content up to 30 wt-%, and then almost stagnated at higher levels of lignin content. As for the oPE-*g*-lignin/LDPE biocomposites, the tensile strength slightly decreased when the oPE-*g*-lignin content was incremented from 10 to 20 wt-%; however, when further increasing the oPE-*g*-lignin content, the tensile strength accordingly rose. The descending tendency of the tensile strength for the pristine-lignin/LDPE biocomposites may be explained by the aforementioned poor interfacial adhesion between the pristine lignin particles and the LDPE matrix. In contrast, for the oPE-*g*-lignin/LDPE biocomposites, the tendency of the tensile strength exhibited a concave upward parabola shape, with the vertex at an oPE-*g*-lignin content of 20 wt-%. In this case, the descending tendency may be attributed to the insufficient interfacial adhesion between the oPE-*g*-lignin particles and the LDPE matrix, while the increasing tendency might have resulted from an increased reinforcing effect of higher-strength (compared with pristine lignin) oPE-*g*-lignin, which should have compensated for such an insufficient interfacial adhesion [21].

On the other hand, the tensile modulus of the pristine-lignin/LDPE biocomposites increased progressively along with the lignin content, and that of the oPE-*g*-lignin/LDPE biocomposites showed a similar tendency. These results are rather common due to the incorporation of stiff lignin particles into the LDPE matrix; indeed, the pristine lignin- and oPE-*g*-lignin-based LDPE biocomposites did not differ much in terms of tensile modulus, indicating that the chemical modification of the lignin’s surface was not enough to enhance it.

Meanwhile, the elongation at the break was drastically reduced for both LDPE biocomposite types when increasing the lignin content. Based on a previous study [46], this tendency could be explained by the lignin characteristically exhibiting less distortion than LDPE when the biocomposite was elongating during the tensile test. These results demonstrate that the chemical modification of pristine lignin with oPE can effectively improve the interfacial adhesion between the lignin particles and the LDPE matrix to enhance the mechanical properties of the resultant LDPE biocomposites.

To investigate the cellular structure of lignin/LDPE biocomposite foams, their cryofractured surfaces were observed by SEM (Figure 4), and their foam characteristics are summarized in Table 2. The SEM micrograph of the neat LDPE foam showed typical polygonal geometries, consistent with previous reports [3,4]. For the pristine-lignin/LDPE biocomposite foams, only the one containing 10 wt-% pristine lignin exhibited cell and size morphologies similar to those of the neat LDPE foam; at a higher lignin content, the foam quality decreased drastically, resulting in a smaller cell size [Figure 4A–E]. In contrast, when the LDPE biocomposites contained 30 wt-% or less oPE-*g*-lignin, their foam quality was similar to that of the neat LDPE foam, as evidenced by the similar cell sizes. However, when the oPE-*g*-lignin content was further increased, the foam quality slightly decreased but was still better than that of the LDPE biocomposites’ foams containing 20 wt-% or more pristine lignin [Figure 4a–e]. All the physical characteristics (Table 2) of the two lignin-reinforced LDPE biocomposite foam series were reduced when increasing the lignin content, but the decreasing tendency was less stiff for the oPE-*g*-linin/LDPE biocomposite foams; surprisingly, the LDPE biocomposite foam containing less than 20 wt-% oPE-*g*-lignin showed almost similar physical characteristics with those of the neat LDPE foam.

To more quantitatively investigate the cellular structure of lignin-reinforced LDPE biocomposite foams, their cell size distribution was analyzed. As shown in Figure 5, the median of the normal distribution curves for all the pristine-lignin/LDPE biocomposite foams shifted toward smaller values compared with the neat LDPE foam; this indicates the presence of numerous cells smaller than the average cell diameter, resulting in a decreased average cell diameter and an increased cell density. The oPE-*g*-lignin/LDPE biocomposite foams exhibited similar bell-shaped normal distribution curves and medians when the oPE-*g*-lignin content was 20 wt-% or lower, as evidenced by the similarity with the curve of the neat LDPE foam. At higher oPE-*g*-lignin contents, the median shifted toward smaller cell sizes.

The cellular structure of polymeric composite-based foams should vary depending on the type, size, and content of the filler [53]. The filler particles incorporated in the polymer matrix can provide more nucleation sites, resulting in the formation of a new cellular structure in the composite. Based on this hypothesis, the lignin/polymer interfacial morphology may significantly influence the cellular structure during the cell nucleation step in the foaming process. When pristine lignin is introduced into LDPE biocomposites, the formation of several microcracks can be expected due to the poor interfacial adhesion between the LDPE matrix and the lignin particles; then, the gas formed by the blowing agent may migrate toward the microcracks, reducing the driving force to increase the size of the initially formed cells and resulting in a smaller average cell diameter.

The volume expansion ratio of the lignin-reinforced LDPE biocomposite foams was also calculated (Figure 6). A high expansion ratio (~17) was observed for the neat LDPE foam. When increasing the pristine lignin content up to 30 wt-%, the expansion ratio gradually decreased and stagnated at higher contents. As for the oPE-*g*-lignin, its introduction did not substantially change the expansion ratio compared with the neat LDPE foam for contents up to 20 wt-%; however, at higher oPE-*g*-lignin contents, the expansion ratio was consistently reduced when increasing the lignin content. This tendency variation might be explained by the improved interfacial adhesion between the lignin particles and the LDPE matrix, which could have minimized the formation of microcracks in the heterogeneous interfacial zone of the biocomposite during the foaming process. Thus, this improved interface adhesion could quietly reduce the gas release from the cells while avoiding cell coalescence.

## 4. Conclusions

Using a typical radical reaction, we demonstrated the feasibility of chemically modifying the surface of lignin grafted to oPE. Through the two-step modification reaction of pristine lignin, oligomeric-polyethylene-grafted lignin was successfully constructed and used to fabricate LDPE biocomposites reinforced with a high-content lignin with a 50 wt-%. The improved interfacial adhesion between the oPE-*g*-lignin particles and the LDPE matrix improved the tensile performance compared with the pristine-lignin/LDPE biocomposites. We also concluded that oPE-*g*-lignin could serve as both a reinforcing filler in the LDPE matrix and a nucleating agent for the recrystallization of the LDPE polymer chain.

The foamability of the lignin-reinforced LDPE biocomposites was evaluated in the presence of azodicarbonamide and dicumyl peroxide. The morphological results showed that the foaming of the lignin-reinforced LDPE biocomposites was not interrupted even though oPE-*g*-lignin was added as high as 20 wt-% and had a quietly uniform cell structure resulting in the foam expansion ratios 17.69 ± 0.92, which are similar to those of neat LDPE foam (17.04 ± 0.44). In contrast, the pristine-lignin/LDPE biocomposites exhibited a poor foam quality at any lignin content. Thus, the usage of oPE-*g*-lignin in lignin-reinforced LDPE biocomposite foams might help to construct more stable and uniformed cellular structures even at a high lignin content compared to the pristine-lignin/LDPE biocomposite foams. The oPE-*g*-lignin/LDPE biocomposite foams with a high lignin content will serve as a reasonable eco-friendly alternative to petroleum-based cushioning packaging materials for electronics, automobiles, and other fields.

## Data Availability

Not applicable.

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
