# Peer review of "Construction, Physical Properties and Foaming Behavior of High-Content Lignin Reinforced Low-Density Polyethylene Biocomposites"

_polymers, 2022, doi:10.3390/polym14132688_

Round 1

Reviewer 1 Report

In the present manuscript entitled “Construction, physical properties and foaming behavior of high-content lignin reinforced low-density polyethylene bio-composites” the authors described the preparation of bio-composites with lignin and LDPE; and studied their foaming behavior. The work is interesting and within the scope of this journal. However there are some major flaws, hence I suggest its rejection or a major revision.

My specific comments are

1.     The authors must emphasize the novelty of the work in the abstract.

2.     The authors did similar work previously. What is the motivation behind this work, please give explanation.

3.     The clear evidence for bio-composites formation should be provided.

4.     The authors should compare the physical properties and foaming behavior with literature.

5.     What are the concentrations of crosslinking agent and blowing agent?

6.     What is mean by DMAc?

7.     What is the role of triethylamine and NaHCo3 in chemical modification of lignin? Why authors used “ionized water” for washing?

8.     The preparation process of modified lignin is confusing. “The reaction mixture was stirred in ice bath and at specific temperature, again did same”.. why?

9.     The authors said melt blending method was used for Lignin-LDPE bio-composites preparation. The explanation is not clear.

10.  What is the role of ZnO in foaming experiment?

11.  In 175 line the author should correct the Figure 1 (SEM) to Figure 2.

12.  Expansion ratio for neat LDPE showing above “16” in figure, but authors wrote “~14” in text. The author should correct it.

13.  The authors wrote abbreviation of oPE-g-lignin and LDPE several times in the manuscript. The author should check it.

Reviewer 2 Report

Comments to authors are listed  below:

The abstract lacks to present the numerical values from significant findings in this paper.

·       The introduction should be extended with additional information, which is related to the application and characterisations of LDPE matrix. Some references below are recommended to strength the quality of this section:

·       https://onlinelibrary.wiley.com/doi/full/10.1002/app.48744

·       https://www.mdpi.com/2073-4360/13/13/2138/htm\

·       https://journals.sagepub.com/doi/full/10.1177/0967391120968441.

·       Please discuss the novelty of your work as there are several numerical and experimental studies in this area.

·       The thermal stability (TGA analysis) should be performed in this paper to improve the quality of this work.  

·       DSC curves should be included besides of Table 1, which summarised the results obtained  from DSC curves.

·       Stress-strain curves should be included to verify the results presented in Figure 3.  

  • The discussion of results is tremendously poor and brief. So, further information should be included by comparing the significant values from the results presented in this paper. 

Round 2

Reviewer 2 Report

Comments to authors are listed below:

1.       Ref. 32 at the References list,  The last name of author should be corrected “Awad” instead of the wrong one “Award”.

2.       The conclusions should be improved by including some significant numerical values presented in the results section.

Author Response

Comments from the editors and reviewers:

<Reviewer 2>

Comments to authors are listed below:

  1. Ref. 32 at the References list, the last name of author should be corrected “Awad” instead of the wrong one “Award”.
  • We apologized the misspelling in the reference. According to reviewer’s comment, we revised the incorrected name ‘Award’ to ‘Awad’ (line 392).

  1. The conclusions should be improved by including some significant numerical values presented in the results section.
  • According to the reviewer’s comment, we revised the Conclusion part (line 291-301).